# GLOBAL NODE ATTENTIONS VIA ADAPTIVE SPECTRAL FILTERS

## ABSTRACT

Graph neural networks (GNNs) have been extensively studied for prediction tasks on graphs. Most GNNs assume local homophily, i.e., strong similarities in local neighborhoods. This assumption limits the generalizability of GNNs, which has been demonstrated by recent work on disassortative graphs with weak local homophily. In this paper, we argue that GNN's feature aggregation scheme can be made flexible and adaptive to data without the assumption of local homophily. To demonstrate, we propose a GNN model with a global self-attention mechanism defined using learnable spectral filters, which can attend to any nodes, regardless of distance. We evaluated the proposed model on node classification tasks over seven benchmark datasets. The proposed model has been shown to generalize well to both assortative and disassortative graphs. Further, it outperforms all state-of-the-art baselines on disassortative graphs and performs comparably with them on assortative graphs.

## 1 INTRODUCTION

Graph neural networks (GNNs) have recently demonstrated great power in graph-related learning tasks, such as node classification (Kipf & Welling, 2017), link prediction (Zhang & Chen, 2018) and graph classification (Lee et al., 2018). Most GNNs follow a message-passing architecture where, in each GNN layer, a node aggregates information from its direct neighbors indifferently. In this architecture, information from long-distance nodes is propagated and aggregated by stacking multiple GNN layers together (Kipf & Welling, 2017; Velickovic et al., 2018; Defferrard et al., 2016). However, this architecture underlies the assumption of local homophily, i.e. proximity of similar nodes. While this assumption seems reasonable and helps to achieve good prediction results on graphs with strong local homophily, such as citation networks and community networks (Pei et al., 2020), it limits GNNs' generalizability. Particularly, determining whether a graph has strong local homophily or not is a challenge by itself. Furthermore, strong and weak local homophily can both exhibit in different parts of a graph, which makes a learning task more challenging.

Pei et al. (2020) proposed a metric to measure local node homophily based on how many neighbors of a node are from the same class. Using this metric, they categorized graphs as assortative (strong local homophily) or disassortative (weak local homophily), and showed that classical GNNs such as GCN (Kipf & Welling, 2017) and GAT (Velickovic et al., 2018) perform poorly on disassortative graphs. Liu et al. (2020) further showed that GCN and GAT are outperformed by a simple multi-layer perceptron (MLP) in node classification tasks on disassortative graphs. This is because the naive local aggregation of homophilic models brings in more noise than useful information for such graphs. These findings indicate that these GNN models perform sub-optimally when the fundamental assumption of local homophily does not hold.

Based on the above observation, we argue that a well-generalized GNN should perform well on graphs, regardless of their local homophily. Furthermore, since a real-world graph can exhibit both strong and weak homophily in different node neighborhoods, a powerful GNN model should be able to aggregate node features using different strategies accordingly. For instance, in disassortative graphs where a node shares no similarity with any of its direct neighbors, such a GNN model should be able to ignore direct neighbors and reach farther to find similar nodes, or at least, resort to the node's attributes to make a prediction. Since the validity of the assumption about local homophily is often unknown, such aggregation strategies should be learned from data rather than decided upfront.

To circumvent this issue, in this paper, we propose a novel GNN model with global self-attention mechanism, called GNAN. Most existing attention-based aggregation architectures perform self-attention to the local neighborhood of a node (Velickovic et al., 2018), which may add local noises in aggregation. Unlike these works, we aim to design an aggregation method that can gather informative features from both close and far-distant nodes. To achieve this, we employ graph wavelets under a relaxed condition of localization, which enables us to learn attention weights for nodes in the spectral domain. In doing so, the model can effectively capture not only local information but also global structure into node representations.

To further improve the generalizability of our model, instead of using predefined spectral kernels, we propose to use multi-layer perceptrons (MLP) to learn the desired spectral filters without limiting their shapes. Existing works on graph wavelet transform choose wavelet filters heuristically, such as heat kernel, wave kernel and personalized page rank kernel (Klicpera et al., 2019b; Xu et al., 2019; Klicpera et al., 2019a). They are mostly low-pass filters, which means that these models implicitly treat high-frequency components as "noises" and have them discarded (Shuman et al., 2013; Hammond et al., 2011; Chang et al., 2020). However, this may hinder the generalizability of models since high-frequency components can carry meaningful information about local discontinuities, as analyzed in (Shuman et al., 2013). Our model overcomes these limitations by incorporating fully learnable spectral filters into the proposed global self-attention mechanism.

From a computational perspective, learning global self-attention may impose high computational overhead, particularly when graphs are large. We alleviate this problem from two aspects. First, we sparsify nodes according to their wavelet coefficients, which enables attention weights to be distributed across the graph sparsely. Second, we observed that spectral filters learned by different MLPs tend to converge to be of similar shapes. Thus, we use a single MLP to reduce redundancy among filters, where each dimension in the output corresponds to one learnable spectral filter. In addition to these, following (Xu et al., 2019; Klicpera et al., 2019b), we use a fast algorithm to efficiently approximate graph wavelet transform, which has computational complexity $O(p \times |E|)$, where $p$ is the order of Chebyshev polynomials and $|E|$ is the number of edges in a graph.

To summarize, the main contributions of this work are as follows:

1. We propose a generalized GNN model which performs well on both assortative and disassortative graphs, regardless of local node homophily.

2. We exhibit that GNN's aggregation strategy can be trained via a fully learnable spectral filter, thereby enabling feature aggregation from both close and far nodes.

3. We show that, unlike commonly understood, higher-frequency on disassortative graphs provides meaningful information that helps improving prediction performance.

We conduct extensive experiments to compare GNAN with well-known baselines on node classification tasks. The experimental results show that GNAN significantly outperforms the state-of-the-art methods on disassortative graphs where local node homophily is weak, and performs comparably with the state-of-the-art methods on assortative graphs where local node homophily is strong. This empirically verifies that GNAN is a general model for learning on different types of graphs.

## 2 PRELIMINARIES

Let $\mathcal{G} = (V, E, A, \boldsymbol{x})$ be an undirected graph with $N$ nodes, where $V$, $E$, and $A$ are the node set, edge set, and adjacency matrix of $\mathcal{G}$, respectively, and $\boldsymbol{x} : V \mapsto \mathbb{R}^m$ is a graph signal function that associates each node with a feature vector. The normalized Laplacian matrix of $\mathcal{G}$ is defined as $\boldsymbol{L} = \boldsymbol{I} - \boldsymbol{D}^{-1/2} \boldsymbol{A} \boldsymbol{D}^{-1/2}$, where $\boldsymbol{D} \in \mathbb{R}^{N \times N}$ is the diagonal degree matrix of $\mathcal{G}$. In spectral graph theory, the eigenvalues $\Lambda = \mathrm{diag}(\lambda_1, ..., \lambda_N)$ and eigenvectors $\boldsymbol{U}$ of $\boldsymbol{L} = \boldsymbol{U} \Lambda \boldsymbol{U}^H$ are known as the graph's spectrum and spectral basis, respectively, where $\boldsymbol{U}^H$ is the Hermitian transpose of $\boldsymbol{U}$. The graph Fourier transform of $\boldsymbol{x}$ is $\hat{\boldsymbol{x}} = \boldsymbol{U}^H \boldsymbol{x}$ and its inverse is $\boldsymbol{x} = \boldsymbol{U} \hat{\boldsymbol{x}}$.

The spectrum and spectral basis carry important information on the connectivity of a graph (Shuman et al., 2013). Intuitively, lower frequencies correspond to global and smooth information on the graph, while higher frequencies correspond to local information, discontinuities and possible noise (Shuman et al., 2013). One can apply a spectral filter $g$ as in Equation 1 and use graph Fourier transform to manipulate signals on a graph in various ways, such as smoothing and denoising (Schaub &

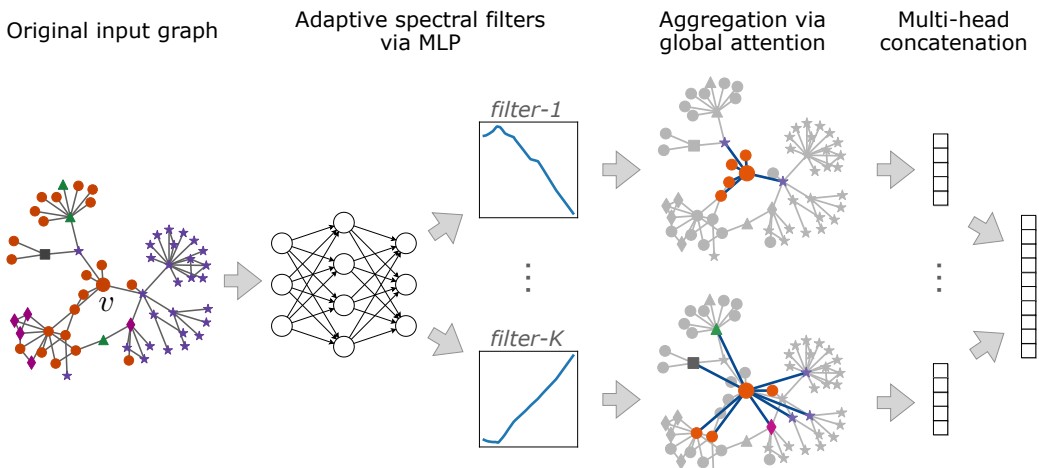

Figure 1: Illustration of a spectral node attention layer on a 3-hop ego network of the central node $v$ from the CITESEER dataset. Node classes are indicated by shape and color. Passing the graph through two learned spectral filters (adaptive spectral filters) place attention scores on nodes, including node $v$ itself. Nodes with positive attention scores are presented in color. Node features are aggregated for node $v$ according to attention scores (aggregation via global attention). The low-pass filter attend to local neighbors (filter 1), while the high-pass filter skips the first hop and attend the nodes in the second hop (filter $K$). The resulting embeddings from multiple heads are then concatenated before being sent to the next layer (multi-head concatenation). Note that we have visualized learned filters from experiments.

Segarra, 2018), abnormally detection (Miller et al., 2011) and clustering (Wai et al., 2018). Spectral convolutions on graphs is defined as the multiplication of a signal $\boldsymbol{x}$ with a filter $g(\Lambda)$ in the Fourier domain, i.e.

$$g(\boldsymbol{L})\boldsymbol{x} = g(\boldsymbol{U}\Lambda\boldsymbol{U}^H)\boldsymbol{x} = \boldsymbol{U}g(\Lambda)\boldsymbol{U}^H\boldsymbol{x} = \boldsymbol{U}g(\Lambda)\hat{\boldsymbol{x}}. \tag{1}$$

When a spectral filter is parameterized by a scale factor, which controls the radius of neighbourhood aggregation, Equation 1 is also known as the Spectral Graph Wavelet Transform (SGWT) (Hammond et al., 2011; Shuman et al., 2013). For example, Xu et al. (2019) uses a small scale parameter $s < 2$ for a heat kernel, $g(s\lambda) = e^{-\lambda s}$, to localize the wavelet at a node.

## 3 PROPOSED APPROACH

Graph neural networks (GNNs) learn lower-dimensional embeddings of nodes from graph structured data. In general, given a node, GNNs iteratively aggregate information from its neighbor nodes, and then combine the aggregated information with its own information. An embedding of node $v$ at the $k$th layer of GNN is typically formulated as

$$\boldsymbol{m}_v = \text{aggregate}(\{\boldsymbol{h}_u^{(k-1)} | u \in \mathcal{N}_v\})$$
$$\boldsymbol{h}_v^{(k)} = \text{combine}(\boldsymbol{h}_v^{(k-1)}, \boldsymbol{m}_v),$$

where $\mathcal{N}_v$ is the set of neighbor nodes of node $v$, $\boldsymbol{m}_v$ is the aggregated information from the neighbors, and $\boldsymbol{h}_v^{(k)}$ is the embedding of the node $v$ at the $k$th layer ($\boldsymbol{h}_v^{(0)} = \boldsymbol{x}_v$). The embedding $\boldsymbol{h}_v^n$ of the node $v$ at the final layer is then used for some prediction tasks. In most GNNs, $\mathcal{N}_v$ is restricted to a set of one-hop neighbors of node $v$. Therefore, one needs to stack multiple aggregation layers in order to collect the information from more than one-hop neighborhood within this architecture.

**Adaptive spectral filters.** Instead of stacking multiple aggregation layers, we introduce a spectral attention layer that rewires a graph based on spectral graph wavelets. A spectral graph wavelet $\boldsymbol{\psi}_v$ at node $v$ is a modulation in the spectral domain of signals centered around the node $v$, given by an $N$-dimensional vector

$$\boldsymbol{\psi}_v = \boldsymbol{U}g(\Lambda)\boldsymbol{U}^H\delta_v, \tag{2}$$

where $g(\cdot)$ is a spectral filter and $\delta_v$ is a one-hot vector for node $v$.

The common choice of a spectral filter is heat kernel. A wavelet coefficient $\psi_{vu}$ computed from a heat kernel can be interpreted as the amount of energy that node $v$ has received from node $u$ in its local neighborhood. In this work, instead of using pre-defined localized kernels, we use multi-layer perceptrons (MLP) to learn spectral filters. With learnable spectral kernels, we obtain wavelet coefficients

$$\boldsymbol{\psi}_v = \boldsymbol{U}\mathrm{diag}(\mathrm{MLP}(\Lambda))\boldsymbol{U}^H \delta_v. \tag{3}$$

Similar to that of a heat kernel, the wavelet coefficient with a learnable spectral filter $\psi_{vu}$ can be understood as the amount of energy that is distributed from node $v$ to node $u$, under the conditions regulated by the spectral filter. Note that we use the terminology wavelet and spectral filter interchangeably as we have relaxed the wavelet definition from (Hammond et al., 2011) so that learnable spectral filters in our work are not necessarily localized in the spectral and spatial domains. Equation 3 requires the eigen-decomposition of a Laplacian matrix, which is expensive and infeasible for large graphs. We follow Xu et al. (2019); Klicpera et al. (2019b) to approximate graph wavelet transform using Chebyshev polynomials (Shuman et al., 2013) (see Appendix A for details).

**Global self-attention.** Unlike the previous work (Xu et al., 2019) where wavelet coefficients are directly used to compute node embeddings, we normalize wavelet coefficients through a softmax layer

$$\boldsymbol{a}_v = \mathrm{softmax}(\boldsymbol{\psi}_v),$$

where $\boldsymbol{a}_v \in \mathbb{R}^N$ is an attention weight vector. With attention weights, an update layer is then formalized as

$$\boldsymbol{h}_v^{(k)} = \sigma \left( \sum_{u=1}^N a_{vu}\boldsymbol{h}_u^{(k-1)}\boldsymbol{W}^{(k)} \right), \tag{4}$$

where $\boldsymbol{W}^{(k)}$ is a weight matrix shared across all nodes in the $k$th layer and $\sigma$ is ELU nonlinear activation. Unlike heat kernel, the wavelet coefficient with a learnable spectral kernel is not localized. Hence, our work can actively aggregate information from far-distant nodes. Note that the update layer is not divided into aggregation and combine steps in our work. Instead, we compute the attention $a_{vv}$ directly from a spectral filter.

**Sparsified node attentions.** With predefined localized spectral filters such as a heat kernel, most of wavelet coefficients are zero due to their locality. In our work, spectral filters are fully learned from data, and consequently attention weights obtained from learnable spectral filters do not impose any sparsity. This means that to perform an aggregation operation we need to retrieve all possible nodes in a graph, which is time consuming with large graphs. From our experiments, we observe that most of attention weights are negligible after softmax. Thus, we consider two sparsification techniques:

1. Discard the entries of wavelet bases that are below a threshold $t$, i.e.

$$\bar{\psi}_{vu} = \begin{cases} \psi_{vu} & \text{if } \psi_{vu} > t \\ -\infty & \text{otherwise.} \end{cases} \tag{5}$$

   The threshold $t$ can be easily applied on all entries of wavelet bases. However, it offers little guarantee on attention sparsity since attention weights may vary, depending on the learning process of spectral filters and the characteristics of different datasets, as will be further discussed in Section 4.2.

2. Keep only the largest $k$ entries of wavelet bases for each node, i.e.

$$\bar{\psi}_{vu} = \begin{cases} \psi_{vu} & \text{if } \psi_{vu} \in \mathrm{topK}(\{\psi_{v0}, ..., \psi_{vN}\}, k) \\ -\infty & \text{otherwise,} \end{cases} \tag{6}$$

   where topK is a partial sorting function that returns the largest $k$ entries from a set of wavelet bases $\{\psi_{v0}, ..., \psi_{vN}\}$. This technique guarantees attention sparsity such that the embedding of each node can be aggregated from at most $k$ other nodes. However, it takes more computational overhead to sort entries since topK has a time complexity of $O(N + k \log N)$.

Table 1: Dataset statistics. We categorize the datasets into assortative and disassortative based on their homophily ($\beta$).

| Category | Dataset | #Nodes | #Edges | #Classes | #Features | Density | $\beta$ |
|---|---|---|---|---|---|---|---|
| Assortative | CORA | 2,708 | 5,429 | 7 | 1,433 | 1.44e-3 | 0.83 |
| | CITESEER | 3,327 | 4,732 | 6 | 3,703 | 8.23e-4 | 0.71 |
| | PUBMED | 19,717 | 44,338 | 3 | 500 | 2.28e-4 | 0.79 |
| Disassortative | CORNELL | 183 | 298 | 5 | 1,703 | 1.68e-2 | 0.11 |
| | TEXAS | 183 | 325 | 5 | 1,703 | 1.77e-2 | 0.06 |
| | WISCONSIN | 251 | 515 | 5 | 1,703 | 1.49e-2 | 0.16 |
| | CHAMELEON | 2277 | 36101 | 5 | 2,325 | 1.21e-2 | 0.25 |

The resulting $\bar{\psi}$ from either of the above techniques is then fed into the softmax layer to compute attention weights. The experiments for comparing these techniques will be discussed in Section 4.2.

We adopt multi-head attention to model multiple spectral filters. Each attention head aggregates node information with a different spectral filter, and the aggregated embedding is concatenated before being sent to the next layer. We can allocate an independent MLP for each of attention heads; however, we found independent MLPs tend to learn spectral filters of similar shapes. Hence, we adopt a single MLP: $\mathbb{R}^N \to \mathbb{R}^{N \times M}$, where $M$ is the number of attention heads, and each column of the output corresponds to one adaptive spectral filter.

We name the multi-head spectral attention architecture as a *global node attention network* (GNAN). The design of GNAN is easily generalizable, and many existing GNNs can be expressed as special cases of GNAN (see Appendix D). Figure 1 illustrates how GNAN works with two attention heads learned from the CITESEER dataset. As shown in the illustration, the MLP learns adaptive filters such as low-band pass and high-band pass filters. A low-band pass filter assigns high attention weights in local neighborhoods, while a high-band pass filter assigns high attention weights on far-distant nodes, which cannot be captured by a one-hop aggregation scheme.

## 4 EXPERIMENTS

To evaluate the performance of our proposed model, we conduct experiments on node classification tasks with assortative graph datasets, where the labels of nodes exhibit strong homophily, and disassortative graph datasets, where the local homophily is weak and labels of nodes represent their structural roles. To quantify the assortativeness of graphs, we use the metric $\beta$ introduced by Pei et al. (2020),

$$\beta = \frac{1}{|V|} \sum_{v \in V} \beta_v \quad \text{and} \quad \beta_v = \frac{|\{u \in \mathcal{N}_v | \ell(u) = \ell(v)\}|}{|\mathcal{N}_v|}, \tag{7}$$

where $\ell(v)$ refers to the label of node $v$. $\beta$ measures the homophily of a graph, and $\beta_v$ measures the homophily of node $v$ in the graph. A graph has strong local homophily if $\beta$ is large and vice versa.

### 4.1 EXPERIMENTAL SETUP

**Baseline methods.** We evaluate two variants of GNAN which only differ in the method used for sparsification: one adopts Equation 5 called GNAN-T, and the other adopts Equation 6 called GNAN-K. We compare both variants against 11 benchmark methods: vanilla GCN (Kipf & Welling, 2017) and its simplified version SGC (Wu et al., 2019); two spectral methods, one using the Chebyshev polynomial spectral filters (Defferrard et al., 2016) and the other using the auto-regressive moving average (ARMA) filters (Bianchi et al., 2019); the graph attention model GAT (Velickovic et al., 2018); APPNP which allows adaptive neighbourhood aggregation using personalized page rank (Klicpera et al., 2019a); three sampling-based approaches, GraphSage (Hamilton et al., 2017), FastGCN Chen et al. (2018) and ASGCN (Huang et al., 2018); and Geom-GCN which targets prediction on disassortative graphs (Pei et al., 2020). We also include MLP in the baselines since it performs better than many GNN-based methods on disassortative graphs (Liu et al., 2020).

Table 2: Micro-F1 results for node classification. The proposed model consistently outperforms the other variants of GNN on disassortative graphs and performs comparably on assortative graphs.

| Method | CORA | CITESEER | PUBMED | CORNELL | TEXAS | WISCONSIN | CHAMELEON |
|---|---|---|---|---|---|---|---|
| GCN | $87.4 \pm 0.2$ | $78.5 \pm 0.5$ | $87.8 \pm 0.2$ | $59.2 \pm 3.2$ | $64.1 \pm 4.9$ | $64.1 \pm 6.3$ | $67.6$ |
| Chebyshev | $88.2 \pm 0.2$ | $79.4 \pm 0.4$ | $89.3 \pm 0.3$ | $76.5 \pm 9.4$ | $79.7 \pm 5.0$ | $82.5 \pm 2.8$ | $66.0 \pm 2.3$ |
| ARMA | $85.2 \pm 2.5$ | $76.7 \pm 0.5$ | $86.3 \pm 5.7$ | $74.9 \pm 2.9$ | $82.2 \pm 5.1$ | $78.4 \pm 4.6$ | $62.1 \pm 3.6$ |
| GAT | $87.6 \pm 0.3$ | $77.7 \pm 0.3$ | $83.0 \pm 0.1$ | $58.9 \pm 3.3$ | $60.0 \pm 5.7$ | $62.0 \pm 5.2$ | $64.9$ |
| SGC | $87.2 \pm 0.3$ | $78.8 \pm 0.4$ | $81.1 \pm 0.3$ | $58.1 \pm 4.6$ | $58.9 \pm 6.1$ | $51.8 \pm 5.9$ | $33.7 \pm 3.5$ |
| GraphSAGE | $86.3 \pm 0.6$ | $77.4 \pm 0.5$ | $89.2 \pm 0.5$ | $67.3 \pm 6.9$ | $82.7 \pm 4.8$ | $77.6 \pm 4.6$ | $51.1 \pm 0.5$ |
| APPNP | $\mathbf{88.4} \pm 0.3$ | $77.6 \pm 0.6$ | $86.0 \pm 0.3$ | $70.3 \pm 9.3$ | $79.5 \pm 4.6$ | $81.2 \pm 2.5$ | $45.3 \pm 1.6$ |
| FastGCN | $86.1 \pm 0.4$ | $78.1 \pm 0.7$ | $86.0 \pm 0.8$ | $72.7 \pm 6.8$ | $68.4 \pm 4.6$ | $68.8 \pm 5.4$ | $42.8 \pm 2.4$ |
| ASGCN | $86.8 \pm 0.4$ | $77.9 \pm 0.3$ | $88.0 \pm 0.7$ | $73.0 \pm 6.1$ | $68.1 \pm 7.7$ | $66.3 \pm 6.5$ | $49.5 \pm 1.5$ |
| Geom-GCN | $86.3 \pm 0.3$ | $\mathbf{81.4} \pm 0.3$ | $89.1 \pm 0.1$ | $75.4$ | $73.5$ | $80.4$ | $68.0$ |
| MLP | $72.1 \pm 1.3$ | $74.9 \pm 1.8$ | $88.6 \pm 0.2$ | $81.4 \pm 6.3$ | $79.2 \pm 6.1$ | $82.7 \pm 4.5$ | $48.5$ |
| GNAN-T* | $87.5 \pm 0.5$ | $79.0 \pm 1.1$ | $\mathbf{89.9} \pm 0.9$ | $\mathbf{82.7} \pm 8.3$ | $\mathbf{85.1} \pm 5.7$ | $85.9 \pm 3.8$ | $66.1 \pm 3.1$ |
| GNAN-K* | $87.2 \pm 0.3$ | $79.3 \pm 0.6$ | $88.7 \pm 0.5$ | $81.4 \pm 10.1$ | $82.4 \pm 5.1$ | $\mathbf{86.3} \pm 3.7$ | $\mathbf{68.7} \pm 2.8$ |

**Datasets.** We evaluate our model and the baseline methods on node classification tasks over three citation networks: CORA, CITESEER and PUBMED (Sen et al., 2008), three webgraphs from the WebKB dataset[1]: WISCONSIN, TEXAS and CORNELL, and another webgraph from Wikipedia called CHAMELEON (Rozemberczki et al., 2019). We divide these datasets into two groups, assortative and disassortative, based on their $\beta$. The details of these datasets are summarized in Table 1.

**Hyper-parameter settings.** For the citation networks, we follow the experimental setup for node classification from (Hamilton et al., 2017; Huang et al., 2018; Chen et al., 2018) and report the results averaged on 10 runs. For the webgraphs, we run each model on the 10 splits provided by (Pei et al., 2020) and take the average, where each split uses 60%, 20%, and 20% nodes of each class for training, validation and testing, respectively. The results we report on GCN and GAT are better than Pei et al. (2020) due to converting the graphs to undirected before training [2]. Geom-GCN uses node embeddings pre-trained from different methods such as Isomap (Tenenbaum et al., 2000), Poincare (Nickel & Kiela, 2017) and struc2vec (Ribeiro et al., 2017). We hereby report the best micro-F1 results among all variants for Geom-GCN.

We use the best-performing hyperparameters specified in the original papers of baseline methods. For hyperparameters not specified in the original papers, we use the parameters from (Fey & Lenssen, 2019). We report the test accuracy results from epochs with the smallest validation loss and highest validation accuracy. Early termination is adopted for both validation loss and accuracy, and the training is thus stopped when neither of validation loss and accuracy improve for 100 consecutive epochs. We use a two-layer GNAN where multi-head's filters are learned using a MLP of 2 hidden layers and then approximated by Chebyshev polynomials. Each layer of the MLP consists of a linear function and a ReLU activation. To avoid overfitting, dropout is applied in each GNAN layer on both attention weights and inputs equally.

## 4.2 RESULTS AND DISCUSSION

We use two evaluation metrics to evaluate the performance of node classification tasks: micro-F1 and macro-F1. The results with micro-F1 are summarized in Table 2, and the results with macro-F1 are provided in Table 3 in the appendix. Overall, on assortative citation networks, GNAN performs comparably with the state-of-the-art methods, ranking first on PUBMED and second on CORA and CITESEER in terms of micro-F1 scores. On disassortative graphs, GNAN outperforms all the state-of-the-art methods by a margin of at least 2.4% and MLP by a margin of at least 1.3%. These results indicate that GNAN can learn spectral filters adaptively based on different characteristics of graphs.

Although our model GNAN performs well on both assortative and disassortative graphs, it is unclear how GNAN performs on disassortative nodes whose neighbors are mostly of different classes in an assortative graph. Thus, we report an average classification accuracy on disassortative nodes at different levels of $\beta_v$ in Figure 2 for the assortative graph datasets CITESEER and PUBMED. The

---

[1]http://www.cs.cmu.edu/afs/cs.cmu.edu/project/theo-11/www/wwkb/
[2]https://openreview.net/forum?id=S1e2agrFvS

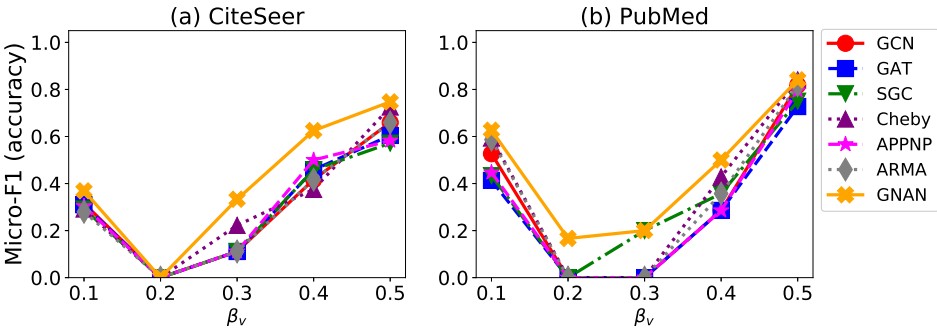

Figure 2: Micro-F1 results for classification accuracy on disassortative nodes ($\beta_v \leq 0.5$). GNAN shows better accuracy on classifying disassortative nodes than the other methods.

nodes are binned into five groups based on $\beta_v$. For example, all nodes with $0.3 < \beta_v \leq 0.4$ belong to the bin at $0.4$. We have excluded CORA from the report since it has very few nodes with low $\beta_v$.

The results in Figure 2 show that all GNNs based on local aggregation schemes perform poorly when $\beta_v$ is low. One may argue that the performance on disassortative graphs might improve by stacking multiple GNN layers together to obtain information from far-distant nodes. However, it turns out that this approach introduces an oversmoothing problem in local aggregation-based GNNs (Li et al., 2018). On the other hand, GNAN outperforms the other GNN-based methods on disassortative nodes, suggesting that adaptive spectral filters reduce local noise in aggregation while allowing far-distant nodes to be attended to.

**Attention sparsification.** The two variants of GNAN use slightly different sparsification techniques to speed up computation. For each node $v$, GNAN-T uses a threshold $t$ to eliminate low $\psi_{vu}$ (Equation 5), thereby sparsifying the resulting attention matrix. However, $t$ cannot control the level of sparsification precisely. In comparison, GNAN-K keeps the $k$ largest $\phi_{vu}$ (Equation 6); therefore it guarantees a certain level of sparsification. Nonetheless, GNAN-K requires a partial sorting which adds an overhead of $O(n + k \log N)$. To further analyze the impact of attention sparsity on runtime, we plot the density of an attention matrix with respect to both $k$ (Figure 3.a and 3.c) and $t$ (Figure 3.b and 3.d). The results are drawn from two datasets: the disassortative dataset CHAMELEON and the assortative dataset CORA. As expected, GNAN-K shows a stable growth in the attention density as the value of $k$ increases. GNAN-T, on the other hand, demonstrates fluctuation in density with $t$ and reaches the lowest density at $t = 1e - 9$ and $t = 1e - 6$ for CORA and CHAMELEON, respectively. We observe that the attention weights tend to converge to similar small values on all nodes when $t$ goes beyond 0.001 in both datasets. To study how efficiency is improved via sparsification, we also plot the training time averaged over 500 epochs in Figure 3. It shows that the model GNAN runs much faster when attention weights are well-sparsified. In our experiments, we find the best results are achieved on $k < 20$ for GNAN-K and $t < 1e - 5$ for GNAN-T. Thus, the model GNAN not only runs faster, but also performs better when attention weights are well-sparsified.

**Frequency range ablation.** To understand how adaptive spectral filters contribute to GNAN's performance on disassortative graphs, we conduct an ablation study on spectral frequency ranges. We first divide the entire frequency range $(0 \sim 2)$ into a set of predefined sub-ranges exclusively, and then manually set the filter frequency responses to zero for each sub-range at a time in order to check the impact of each sub-range on the performance of classification. By doing so, the frequencies within a selected sub-range do not contribute to neither node attention nor feature aggregation, therefore helping to reveal the importance of the sub-range. We consider three different lengths of sub-ranges, i.e., step=1.0, step=0.5, and step=0.25. The results of frequency ablation on the three assortative graphs are summarized in Figure 4. The results for step=1.0 reveal the importance of high-frequency range $(1 \sim 2)$ on node classification of disassortative graphs. The performances are significantly dropped by ablating high-frequency range on all datasets. Further investigation at the finer-level sub-ranges (step=0.5) shows that sub-range $0.5 \sim 1.5$ has the most negative impact on performance, whereas the most important sub-range varies across different datasets at the finest level (step=0.25). This finding matches our intuition that low-pass filters used in GNNs underlie the local node homophily assumption in a similar way as naive local aggregation. We suspect the choice of

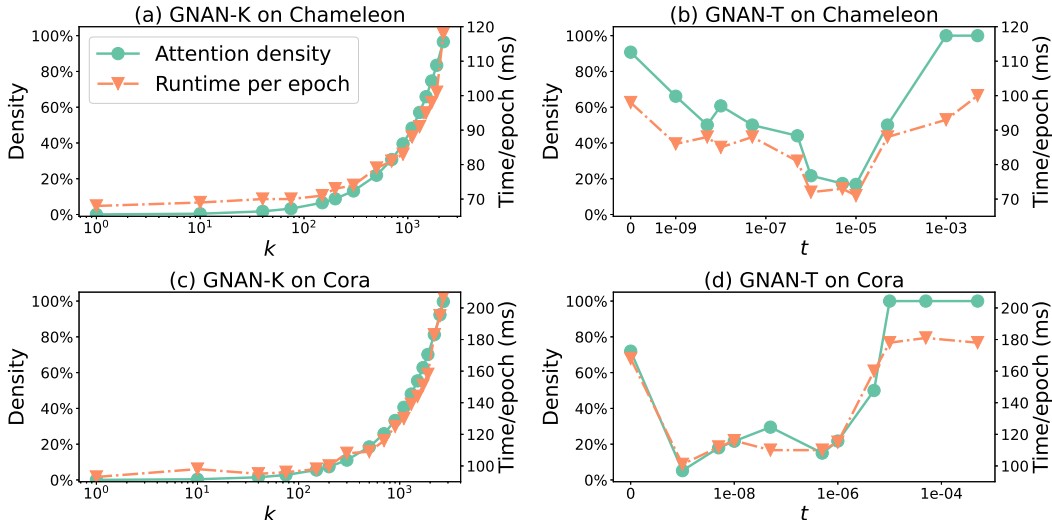

Figure 3: Attention matrix density and training runtime with respect to $k$ and $t$. Attention matrix is effectively sparsified by both $k$ and $t$, which improves runtime efficiency. Note that the density is not monotonically increasing for GNAN-T since the threshold is applied to the "learnable" attention weights. When all values of $\psi$ are below $t$, the density becomes 100% as a result of the softmax normalization.

low-pass filters also relates to oversmoothing issues in spectral methods (Li et al., 2018), but we leave it for future work.

**Attention head ablation.** In GNAN, each head uses a spectral filter to produce attention weights. To delve the importance of a spectral filter, we further follow the ablation method used by (Michel et al., 2019). Specifically, we ablate one or more filters by manually setting their attention weights to zeros. We then measure the impact on performance using micro-F1. If the ablation results in a large decrease in performance, the ablated filters are considered important. We observe that all attention heads (spectral filters) in GNAN are of similar importance, and only all attention heads combined produce the best performance. Please check Appendix C for the detailed results.

## 5 RELATED WORK

Graph neural networks have been extensively studied recently. We categorize work relevant to ours into three perspectives and summarize the key ideas.

**Attention on graphs.** Graph attention networks (GAT) (Velickovic et al., 2018) was the first to introduce attention mechanisms on graphs. GAT assigns different importance scores to local neighbors via attention mechanism. Similar to other GNN variants, long-distance information propagation in GAT is realized by stacking multiple layers together. Therefore, GAT suffers from the oversmoothing issue (Zhao & Akoglu, 2020). Zhang et al. (2020) improve GAT by incorporating both structural and feature similarities while computing attention scores.

**Spectral graph filters and wavelets.** Some GNNs also use graph wavelets to extract information from graphs. Xu et al. (2019) applied graph wavelet transform defined by Shuman et al. (2013) in GNNs. Klicpera et al. (2019b) proposed a general GNN argumentation using graph diffusion kernels to rewire the nodes. Donnat et al. (2018) used heat wavelet to learn node embeddings in unsupervised ways and showed that the learned embeddings closely capture structural similarities between nodes. Other spectral filters used in GNNs can also be viewed as special forms of graph wavelets (Kipf & Welling, 2017; Defferrard et al., 2016; Bianchi et al., 2019). Coincidentally, Chang et al. (2020) also noticed useful information carried by high-frequency components from a graph Laplacian. Similarly, they attempted to utilize such components using node attentions. However, they resorted to the traditional choice of heat kernels and applied such kernels separately to low-frequency

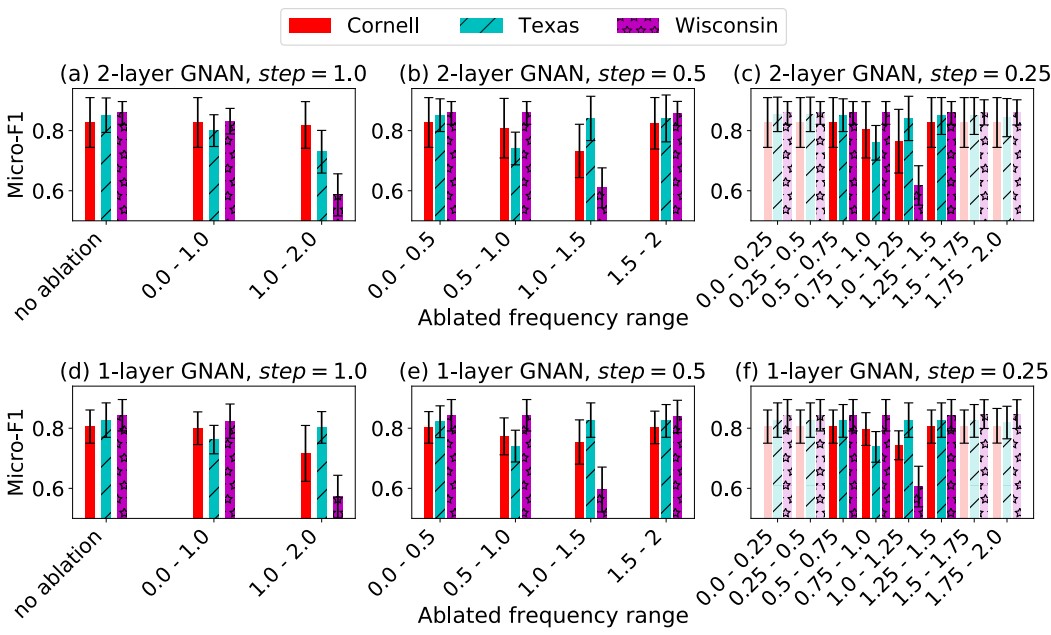

Figure 4: Micro-F1 with respect to an ablated frequency range on disassortative graphs. We divide the frequency range into a set of sub-ranges with different lengths. The results (a) and (d) reveal the importance of high-frequency range $(1 \sim 2)$. Further experiments show that there is a subtle difference in the most important range across datasets, but it ranges between $(0.75 \sim 1.25)$.

and high-frequency components divided by a hyperparameter. In addition to this, their work did not link high-frequency components to disassortative graphs.

**Prediction on disassortative graphs.** Pei et al. (2020) have drawn attention to GCN and GAT's poor performance on disassortative graphs very recently. They tried to address the issue by essentially pivoting feature aggregation to structural neighborhoods from a continuous latent space learned by unsupervised methods. Another attempt to address the issue was proposed by Liu et al. (2020). They proposed to sort locally aggregated node embeddings along a one-dimensional space and used a one-dimensional convolution layer to aggregate embeddings a second time. By doing so, non-local but similar nodes can be attended to.

Although our method shares some similarities in motivation with the aforementioned work, it is fundamentally different in several aspects. To the best of our knowledge, our method is the first to learn spectral filters as part of supervised training on graphs. It is also the first architecture we know that computes node attention weights purely from learned spectral filters. As a result, in contrast to commonly used heat kernel, our method utilizes high-frequency components of a graph, which helps prediction on disassortative graphs.

## 6 CONCLUSION

In this paper, we study the node classification tasks on graphs where local node homophily is weak. We argue the assumption of local homophily is the cause of poor performance on disassortative graphs. In order to design more generalizable GNNs, we suggest that a more flexible and adaptive feature aggregation scheme is needed. To demonstrate, we have introduced the global node attention network (GNAN) which achieves flexible feature aggregation using learnable spectral graph filters. By utilizing the full graph spectrum adaptively via the learned filters, GNAN is able to aggregate features from nodes that are close and far. For node classification tasks, GNAN outperforms all benchmarks on disassortative graphs, and performs comparably on assortative graphs. On assortative graphs, GNAN also performs better for nodes with weak local homophily. Through our analysis, we find the performance gain is closely linked to the higher end of the frequency spectrum.

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

APPENDIX

## A   GRAPH SPECTRAL FILTERING WITHOUT EIGEN-DECOMPOSITION

Chebyshev polynomials approximation has been the de facto approximation method for avoiding eigen-decomposition in spectral graph filters. It has been commonly used in previous works Hammond et al. (2011); Sakiyama et al. (2016); Xu et al. (2019). We hereby use it to approximate Equation 3. In fact, other approximation methods can also be used for the purpose, such as the Jackson-Chebychev polynomials (Napoli et al., 2016) but we will leave it for future study. Briefly, in Chebyshev polynomial approximation, the graph signal filtered by a filter $g(L)$ is approximated as $\tilde{g}(L)$, and represented as a sum of recursive polynomials (Sakiyama et al., 2016):

$$\tilde{g}(\boldsymbol{L})\boldsymbol{x} = \left\{\frac{1}{2}c_0 + \sum_{i=1}^{p} c_i \bar{T}_i(\boldsymbol{L})\right\}\boldsymbol{x} \tag{8}$$

where $\bar{T}_0 = 1, \bar{T}_1(\boldsymbol{L}) = 2(\boldsymbol{L}-1)/\lambda_{max}, \bar{T}_i(\boldsymbol{L}) = 4(\boldsymbol{L}-1)\bar{T}_{i-1}/\lambda_{max} - \bar{T}_{i-2}(\boldsymbol{L})$, and

$$c_i = \frac{2}{S}\sum_{m=1}^{S} cos\Big(\frac{\pi i(m-\frac{1}{2})}{S}\Big) \times g\Big(\frac{\lambda_{max}}{2}\Big(cos\Big(\frac{\pi(m-\frac{1}{2})}{S}+1\Big)\Big)\Big) \tag{9}$$

for $i = 0, ..., p$, where p is the approximation order, S is the number of sampling points and is normally set to $S = p + 1$.

In Equation 3, MLP is used to produce the filter responses so we have $g = $ MLP in Equation 9. The above equation is differentiable so the parameters in MLP can be learned by gradient decent from the loss function. The above approximation has a time complexity of $O(p \times |E|)$, so that the complexity for Equation 3 is also $O(p \times |E|)$. Please note, while Chebyshev polynomials are mentioned in both our method and ChevNet, however they are used in fundamentally different ways: ChevNet uses the simplified Chebyshev polynomials as the polynomial filter directly, while we use it as a method to approximate the filtering operation. Naturally, approximation error reduces while a larger $p$ is used, which is also why we have $p > 12$ in our model.

## B   FURTHER EXPERIMENT RESULTS

We provide the macro-F1 scores on the classification task in Table 3. The proposed model outperforms the other models on disassortative graphs and performs comparable on the assortative graphs.

Table 3: Macro-F1 for node classification task.

| Method | CORA | CITESEER | PUBMED | CORNELL | TEXAS | WISCONSIN | CHAMELEON |
|---|---|---|---|---|---|---|---|
| GCN | $86.0 \pm 0.3$ | $72.0 \pm 1.6$ | $86.8 \pm 0.2$ | $24.1 \pm 9.1$ | $34.0 \pm 5.7$ | $37.6 \pm 9.2$ | − |
| Chebyshev | $\mathbf{86.8} \pm 0.3$ | $74.1 \pm 1.0$ | $88.7 \pm 0.3$ | $53.6 \pm 17.6$ | $64.1 \pm 12.4$ | $52.9 \pm 7.7$ | $65.9 \pm 2.4$ |
| ARMA | $80.2 \pm 6.8$ | $66.4 \pm 0.4$ | $81.6 \pm 13.9$ | $48.5 \pm 9.3$ | $69.1 \pm 12.3$ | $53.3 \pm 7.1$ | $60.7 \pm 6.3$ |
| GAT | $86.4 \pm 0.4$ | $69.2 \pm 1.0$ | $81.6 \pm 0.1$ | $19.0 \pm 2.8$ | $26.5 \pm 6.8$ | $30.0 \pm 5.2$ | − |
| SGC | $86.0 \pm 0.3$ | $74.7 \pm 1.2$ | $79.8 \pm 0.3$ | $21.9 \pm 8.5$ | $23.2 \pm 7.5$ | $35.9 \pm 6.3$ | $31.3 \pm 4.4$ |
| GraphSAGE | $85.2 \pm 0.1$ | $74.2 \pm 0.6$ | $88.7 \pm 0.6$ | $49.2 \pm 10.9$ | $62.9 \pm 9.6$ | $63.7 \pm 12.4$ | $51.6 \pm 0.4$ |
| APPNP | $87.0 \pm 0.4$ | $70.2 \pm 1.4$ | $84.8 \pm 0.3$ | $39.6 \pm 16.6$ | $61.0 \pm 8.8$ | $55.8 \pm 5.7$ | $44.0 \pm 1.8$ |
| FastGCN | $84.5 \pm 0.5$ | $72.0 \pm 2.3$ | $84.6 \pm 0.8$ | $47.6 \pm 13.0$ | $38.9 \pm 7.5$ | $41.8 \pm 10.6$ | $40.3 \pm 3.4$ |
| ASGCN | $85.7 \pm 0.5$ | $\mathbf{75.4} \pm 0.4$ | $86.9 \pm 0.7$ | $54.3 \pm 11.5$ | $45.2 \pm 10.0$ | $43.4 \pm 11.5$ | $48.6 \pm 1.4$ |
| Geom-GCN | $85.1 \pm 0.3$ | $76.9 \pm 0.5$ | $88.5 \pm 0.1$ | − | − | − | − |
| MLP | $67.2 \pm 2.5$ | $67.6 \pm 3.5$ | $88.1 \pm 0.2$ | $63.0 \pm 12.3$ | $61.7 \pm 15.2$ | $54.6 \pm 11.5$ | − |
| GNAN-T* | $86.4 \pm 0.4$ | $74.1 \pm 1.9$ | $\mathbf{89.2} \pm 1.0$ | $\mathbf{66.6} \pm 15.3$ | $\mathbf{72.0} \pm 10.6$ | $\mathbf{63.5} \pm 7.0$ | $65.7 \pm 3.4$ |
| GNAN-K* | $87.2 \pm 0.3$ | $79.3 \pm 0.6$ | $87.9 \pm 0.8$ | $64.3 \pm 20.1$ | $66.3 \pm 13.1$ | $64.6 \pm 6.2$ | $\mathbf{68.5} \pm 2.1$ |

## C   ABLATION STUDY ON FILTERS

We provide the detailed version of Figure 4 (c) and (f) in Figure 5.

We further ablated attention head to check the importance of each head in classification.

**Ablating all but one spectral filter**. In GNAN, each head uses a filter to produce spectral attention weights. To delve the importance of a filter, we follow the ablation method used by (Michel et al., 2019). Specifically, we ablate one or more filters by manually setting the attention scores to zeros. We then measure the impact on performance using micro-F1. If the ablation results in a large decrease in performance, the ablated fitlerbank(s)

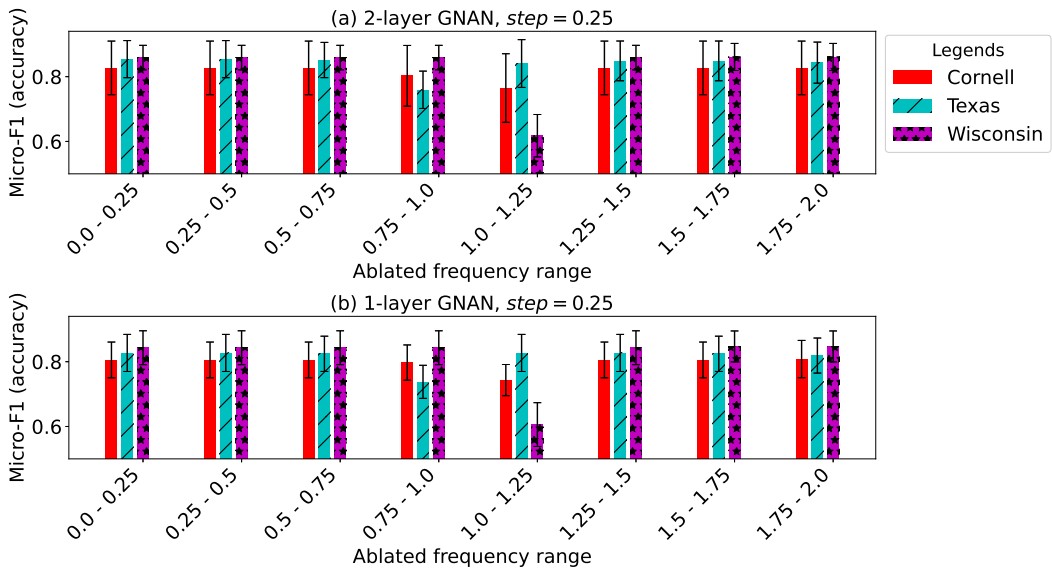

Figure 5: Full details of the performances on frequency ablation at 0.25 level.

is considered important. The results are summarized in Table 4a. All attention head (filters) in GNAN are of similar importance, and only all heads combined produces the best performance.

**Ablating only one spectral filter**. We then examine performance differences by ablating one filter only and keeping all other fitlerbanks Table 4b. Different with above, ablating just one fitlerbank only decreases performance by a small margin. Moreover, ablating some fitlerbanks does not impact prediction performance at all. This is an indicator of potential redundancies in the filters. We leave the redundancy reduction in the model for future work.

Table 4: Ablation study on attention head (filter). We use 12 attention heads for CORA and PUBMED, and 14 heads for CITESEER.

(a) Test accuracy by keeping only one filter

| Dataset \ Head | 1 | 2 | 3 | 4 | 5 | 6 | 7 | 8 | 9 | 10 | 11 | 12 | 13 | 14 |
|---|---|---|---|---|---|---|---|---|---|---|---|---|---|---|
| CORA | -2.10% | -2.10% | -1.30% | -2.10% | -1.50% | -2.10% | -2.10% | -2.10% | -1.40% | -1.50% | -2.10% | -1.30% | - | - |
| CITESEER | -1.8% | -2.0% | -1.7% | -1.7% | -1.8% | -1.8% | -1.7% | -1.9% | -1.7% | -1.9% | -1.7% | -1.7% | -1.9% | -1.9% |
| PUBMED | -4.30% | -4.40% | -4.40% | -5.40% | -4.30% | -5.40% | -5.40% | -5.40% | -4.30% | -4.30% | -4.30% | -5.40% | - | - |

(b) Test accuracy by ablating only one filter

| Dataset \ Head | 1 | 2 | 3 | 4 | 5 | 6 | 7 | 8 | 9 | 10 | 11 | 12 | 13 | 14 |
|---|---|---|---|---|---|---|---|---|---|---|---|---|---|---|
| CORA | 0.00% | 0.00% | -0.30% | 0.00% | -0.40% | 0.00% | 0.00% | 0.00% | -0.40% | -0.40% | 0.00% | -0.40% | - | - |
| CITESEER | -0.20% | -0.30% | -0.70% | -0.70% | -0.40% | -0.60% | -0.80% | 0.00% | -0.60% | 0.00% | -0.80% | -0.50% | 0.00% | 0.00% |
| PUBMED | -0.80% | -0.80% | -0.70% | 0.00% | -0.80% | 0.00% | 0.00% | 0.00% | -0.70% | -0.80% | -0.80% | 0.00% | - | - |

# D    CONNECTIONS TO OTHER METHODS

In this section, we show GNAN has strong connection to existing models, and many GNNs can be expressed as a special case of GNAN under certain conditions.

## D.1    CONNECTION TO GCN

A GCN (Kipf & Welling, 2017) layer can be expressed as

$$h_v^{(k)} = \text{ReLU}(\sum_{u=1}^{N} \hat{a}_{vu} h_u^{(k-1)} W^{(k)})$$

where $\hat{a}_{vu}$ is the elements from the $v$th row of the symmetric adjacency matrix

$$\hat{A} = \tilde{D}^{-1/2} \tilde{A} \tilde{D}^{-1/2} \quad \text{where} \quad \tilde{A} = A + I_N, \ \tilde{D}_{vv} = \sum_{u=1}^{N} \tilde{A}_{vu}$$

So that

$$\hat{a}_{vu} = \begin{cases} 1 & \text{if } e_{vu} \in E \\ 0 & if \ e_{vu} \notin E \end{cases}$$

Therefore, GCN can be viewed as a case of Equation 4 with $\sigma = \text{ReLU}$ and $a_{vu} = \hat{a}_{vu}$

## D.2    CONNECTION TO POLYNOMIAL FILTERS

Polynomial filters localize in a node's $K$-hop neighbors utilizing $K$-order polynomials (Defferrard et al., 2016), most of them takes the following form:

$$g_\theta(\Lambda) = \sum_{k=0}^{K-1} \theta_k \Lambda^k$$

where $\theta_k$ is a learnable polynomial coefficient for each order. Thus a GNN layer using a polynomial filter becomes

$$h_v^{(k)} = \text{ReLU}(\sum_{u=1}^{N} U g_\theta(\Lambda) U^T h_u)$$

which can be expressed using Equation 4 with $W^{(k)} = I_N$, $\sigma = \text{ReLU}$ and $a_{vu} = (U g_\theta(\Lambda) U^T)_{vu}$. In comparison, our method uses a MLP to learn the spectral filters instead of using a polynomial filter. Also, in our method, coefficients after sparsification and normalization are used as directly as attentions.

## D.3    CONNECTION TO GAT

Our method is inspired by and closely related to GAT (Velickovic et al., 2018). To demonstrate the connection, we firstly define a matrix $\Phi$ where each column $\phi_v$ is the transformed feature vector of node $v$ concatenated with feature vector of another node (including node $v$ itself) in the graph.

$$\phi v = ||_{j=0}^{N} [W h_v || W h_u] \tag{10}$$

GAT multiplies each column of $\Phi$ with a learnable weight vector $\alpha$ and masks the result with the adjacency $A$ before feeding it to the nonlinear function LeakyRelu and softmax to calculate attention scores. The masking can be expressed as a Hadamard product with the adjacency matrix $A$ which is the congruent of a graph wavelet transform with the filter $g(\Lambda) = I - \Lambda$:

$$\Psi = A = D^{\frac{1}{2}} U(I - \Lambda) U^T D^{\frac{1}{2}} \tag{11}$$

And the GAT attention vector for node $i$ become

$$a_v = \text{softmax}(\text{LeakyReLU}(\alpha^{\text{T}} \phi_{\text{v}} \odot \bar{\psi}_{\text{v}})) \tag{12}$$

where $\bar{\psi}_v$ is the $v$th row of $\Psi$ after applying Equation 5 with $t = 0$, $\odot$ denotes the Hadamard product, as of (Velickovic et al., 2018).

In comparison with our method, GAT incorporate node features in the attention score calculation, while node attentions in our methods are purely computed from the graph wavelet transform. Also, attentions in GAT are masked by $A$, which means the attentions are restricted to node $v$'s 1-hop neighbours only.

### D.4 CONNECTION TO SKIP-GRAM METHODS

Skip-gram models in natural language processing are shown to be equivalent to a form of matrix factorization (Levy & Goldberg, 2014). Recently Qiu et al. (2018) proved that many Skip-Gram Negative Sampling (SGNS) models used in node embedding, including DeepWalk (Perozzi et al., 2014), LINE (Tang et al., 2015b), PTE (Tang et al., 2015a), and node2vec (Grover & Leskovec, 2016), are essentially factorizing implicit matrices closely related to the normalized graph Laplacian. The implicit matrices can be presented as graph wavelet transforms on the graph Laplacian. For simplicity, we hereby use DeepWalk, a generalized form of LINE and PTE, as an example. Qiu et al. (2018) shows DeepWalk effectively factorizes the matrix

$$\log\left(\frac{\text{vol}(\mathcal{G})}{T}(\sum_{r=1}^{T}\boldsymbol{P}^r)\boldsymbol{D}^{-1}\right) - \log(b) \tag{13}$$

where $\text{vol}(\mathcal{G}) = \sum_v D_{vv}$ is the sum of node degrees, $\boldsymbol{P} = \boldsymbol{D}^{-1}\boldsymbol{A}$ is the random walk matrix, $T$ is the skip-gram window size and $b$ is the parameter for negative sampling. We know

$$\boldsymbol{P} = \boldsymbol{I} - \boldsymbol{D}^{-\frac{1}{2}}\boldsymbol{L}\boldsymbol{D}^{\frac{1}{2}} = \boldsymbol{D}^{-\frac{1}{2}}\boldsymbol{U}(\boldsymbol{I} - \Lambda)\boldsymbol{U}^T\boldsymbol{D}^{\frac{1}{2}}$$

So Equation 13 can be written using graph Laplacian as:

$$\log\left(\frac{\text{vol}(\mathcal{G})}{T}\boldsymbol{D}^{-\frac{1}{2}}\sum_{r=1}^{T}(\boldsymbol{I} - \boldsymbol{L})^r\boldsymbol{D}^{\frac{1}{2}}\right) - \log(b)$$

Or, after eigen-decomposition, as:

$$\boldsymbol{M} = \log\left(\frac{\text{vol}(\mathcal{G})}{Tb}\boldsymbol{D}^{-\frac{1}{2}}\boldsymbol{U}\sum_{r=1}^{T}(\boldsymbol{I} - \Lambda)^r\boldsymbol{U}^T\boldsymbol{D}^{\frac{1}{2}}\right) \tag{14}$$

where $\boldsymbol{U}\sum_{r=1}^{T}(\boldsymbol{I} - \Lambda)^r\boldsymbol{U}^T$, denoted as $\boldsymbol{\psi}_{sg}$, is a wavelet transform with the filter $g_{sg}(\lambda) = \sum_{r=1}^{T}(1 - \lambda)^r$. Therefore, DeepWalk can be seen a special case of Equation 4 where:

$$\boldsymbol{a}_v = \begin{cases} \psi_v & \text{if } v = k \\ 0 & \text{if } v \neq u \end{cases}$$

Assigning $\boldsymbol{H} = \boldsymbol{W} = \boldsymbol{I}$, $K = 1$ and $\sigma(X) = \log(\frac{\text{vol}(\mathcal{G})}{Tb}\boldsymbol{D}^{-\frac{1}{2}}\boldsymbol{X}\boldsymbol{D}^{\frac{1}{2}})$. We have

$$\boldsymbol{h}'_i = \text{FACTORIZE}(\sigma(\boldsymbol{a_i})) \tag{15}$$

where FACTORIZE is a matrix factorization operator of choice. Qiu et al. (2018) uses SVD in a generalized SGNS model, where the decomposed matrix $\boldsymbol{U}_d$ and $\Sigma_d$ from $\boldsymbol{M} = \boldsymbol{U}_d\Sigma\boldsymbol{V}_d$ is used to obtain the node embedding $\boldsymbol{U}_d\sqrt{\Sigma_d}$.

