# OpenReview forum: "Global Node Attentions via Adaptive Spectral Filters"
_ICLR.cc/2021/Conference — Reject_

### Official Review · AnonReviewer1 · 2020-10-25
**High complexity and weak experiments**

**Rating:** 4
**Confidence:** 5

**Review:**

This work propose a new GNN architecture to help GNN break its limitation on only working over homophilic networks. The technical is to use introduce graph global attention.

I think the paper is written okay. The motivation is clear. The solution is reasonable. However, I have following criticisms:
1. This work has limited novelty. Observing that GCN cannot work well over heterophilic networks is not a new idea and observation. Using attention to capture the features from far-away nodes is natural but not novel. I do not think that it is reasonable to argue against other works, e.g. [1] that adopts the above idea by saying they are not expressive enough. Expressiveness sometimes may lead to model overfitting. Actually, ChevNet [2] can also capture far-aways nodes and be expressive enough. Why does it not work well? I guess that it is due to some overfitting issue. Moreover, if I understand it correctly, the limited difference between this work and [3] is most likely the global attention, which has very limited contribution.

2. Although the work claims everywhere to tend to decrease the complexity, when computing the global attention, one still needs to do computation for every pair of nodes, which is of course not scalable for even medium-sized graphs.

3. The heterophilic networks used for evaluation are very small with only several hundred nodes. Why not try larger ones, say actor, Cham. in [4]? I guess the computational issue comes from the global attention.

[1] Non-Local Graph Neural Networks.
[2] Convolutional neural networks on graphs with fast localized spectral filtering.
[3] Graph wavelet neural network
[4] Geom-gcn: Geometric graph convolutional networks.


---post-discussion update----
I would like to thank the authors for preparing the rebuttal and attending our discussion. However, I still think the complexity is a concern of this work. I do not think that Eq. (3) can be implemented within the complexity that the authors claimed. Moreover, if the authors use another way to compute the attention scores, that way should be very clearly stated instead of written in a different form. Given the high complexity, I cannot clearly see the advantage of this work in comparison to [1], as the non-local attention has been proposed in [1] already.

[1] Non-Local Graph Neural Networks.

---

> ### Author Response · Authors · 2020-11-23
> **Response to AnonReviewer #1**
>
> We greatly appreciate your helpful comments, and hereby address your concerns as follows:
>
> ### (1) Novelty
>
> The novelty of our model, besides the global attention you have mentioned, lies in the design of "learning" spectral filters. This learning ability on spectral filters empowers our model to adaptively discover a combination of low-frequency filters and high-frequency filters for learning meaningful node representations, without making any prior assumption on local homophily. This is also our key observation in this work that leads to overcoming the issue that existing GNNs cannot work well over heterophilic networks. In this regard, we do not claim that the novelty of our work is to observe this issue or use attention; instead, the novelty of our work is to solve this issue by designing fully learnable spectral filters without compromising computational efficiency. This is how our work differs from the existing work, including the work in [3] which pre-defines wavelet filters heuristically.
>
> Similarly, for ChevNet [2], it is designed to localize within local node neighbourhoods, where the range of node neighbourhoods is determined by a hyper-parameter that is usually small. While ChevNet can reach far-away nodes using a large $K$, all nodes within the $K$ hop are also captured, resulting in degraded performance on heterophilic networks. Thus, this design restricts the generalizability of ChevNet on graphs when the assumption of local homophily does not hold.
>
> In addition, we would like to thank the reviewer for pointing out the misuse of expressiveness. We have fixed the related sentence in Section 1.
>
>
> ### (2) Complexity
>
> Our work does not require computation for each node pairs. This is because we compute attention weights based on graph wavelet transform, which is approximated using Chebyshev polynomials (same trick used in [3]). Thus, our model has the complexity $O(m\times |E|)$, where $|E|$ is the number of edges and $m$ is the order of Chebyshev polynomials. For comparison with the other methods targeting heterophilic networks, the complexity of the method in [4] is $O(N^2)$ and in [1] is $O(Nlog(N))$ where $N$ is the number of nodes. For real-world graphs, $|E|$ is often much smaller than $N^2$.
>
> Regarding attention sparsification, our model has introduced two sparsification techniques (please refer to the revised paper for technical details in Section 3 and related discussions on experiments in Section 4.2). Nonetheless, attention sparsification does not change the computational complexity of our model since the computational complexity measures the worst-case computational cost, not the actual computational cost. The purpose of sparsifying attentions in our work is to reduce the actual computational cost using a sparse attention matrix. We have added Figure 3 in the revised paper to illustrate how attention sparsification can help reduce the computational cost and thus improve runtime efficiency in our model.
>
>
> ### (3) Datasets
>
> We have added experimental results for Chameleon (2,277 nodes and 36,101 edges) in our revised paper. We are also running experiments on another two datasets Actor (7,600 nodes and 33,544 edges) and Squirrel (5,201 nodes and 217,073 edges) from [4] and will report the results once they are available. Also, it might be worth mentioning that, among all the datasets in our experiments including Chameleon, Actor and Squirrel, PubMed is the largest dataset with 19,717 nodes and 44,338 edges from a computational perspective.
>
> \
> We hope our responses have resolved your questions. Please let us know if you need any further clarification regarding our paper, and we hope you can re-evaluate our paper based on our responses.
>
>
> [1] Meng Liu, Zhengyang Wang, and Shuiwang Ji. "Non-Local Graph Neural Networks", 2020 \
> [2] Micha ̈el Defferrard, Xavier Bresson, and Pierre Vandergheynst. Convolutional Neural Networks on Graphs with Fast Localized Spectral Filtering" 2016 \
> [3] Bingbing Xu et al. “Graph Wavelet Neural Network”. 2019 \
> [4] Hongbin  Pei  et  al.  “Geom-GCN:  Geometric  Graph  Convolutional  Net-works”. 2020

---

> > ### Comment · AnonReviewer1 · 2020-11-23
> > **Further comments**
> >
> > Many thanks for the authors' detailed response. I have a few further questions.
> > ## Novelty
> > Regarding the issue of ChevNet, I do not agree with the authors. By setting a large K (actually K does not need to be very large due to the small world property of networks, K:10~20), ChevNet is able to incorporate the entire graph. However, I do not think the argument "all nodes within the  hop are also captured, resulting in degraded performance on heterophilic networks" is correct. If it may indeed learn a high pass filter, ChevNet will not perform only smoothing the local nodes and can also emphasize nodes that are far away. The failure of ChevNet is actually over-overparameterization and overfitting. Regarding this point, I still cannot see the clear contribution from non-local attention.
> >
> > ## Complexity
> > I do not think the argument of authors is correct. Before the model performs top-K or any other attention sparsification, the model first needs to compute the entire \phi matrix. I did not see any way to make it linear in O(V). Eq.(3) needs to first compute full matrix eigen decomposition, which is with complexity already more than w(|V|^2) itself. Actually, the essence of non-local attention also makes the model unable to leverage efficient matrix eigen decomposition because the non locality means the entire range of eigenvalues are needed.
> >
> > ## Datasets
> > I also agree with AC1. The full comparison of the algorithmic complexity should be provided. The complexity of Eq.3 should also be provided.
> >
> > Regarding this, I still do not think the paper achieves the bar of ICLR.

---

> > > ### Author Response · Authors · 2020-11-25
> > > **Response to AnonReviewer1 further comments**
> > >
> > > Thanks again for your comments. We are happy to discuss the questions further.
> > >
> > > ### Novelty
> > >
> > > We agree with you that, when K is large enough, ChevNet is able to incorporate the entire graph. However, as reported by Kipf and Welling in their work for GCN [5], simplified Chebyshev filters by restricting to 1-hop neighborhood (K=1) often provide better performance on graphs than the cases with larger K. Further, we believe ChevNet remains to be a localized method that resembles a low-pass diffusion filter in practice. A graph laplaian $L$ is closely related to random walks, and its power $L^k$ can be seen as a form of the stationary probabilistic state of step-$k$ random walks. A Chebyshev polynomial filter $g_\theta(L)=\sum_{k=0}^{K-1}\theta_kL^k$ is essentially a sum of weighted random walk states where the step $k\in \{0, ..., K-1\}$. Thus, the embedding of a node is a sum of features from its K-hop neighbours multiplied by a $k$-step random-walk probability matrix, and weighted by $\theta_k$. Therefore, Chebyshev polynomial filters work similarly as pagerank kernels as shown by [6] which has also been shown to be closely related to heat kernels[7]. In fact, a recent work has shown Chebyshev polynomial filters are equivalent to graph diffusion (GDC)[8] which is based on the assumption of homophily, i.e. 'birds of a feather flock together', as reported by the authors.
> > >
> > > In comparison, our method uses MLP which is not restricted by the aforementioned constraints, and thus is more adaptive. On one hand, we observed MLP converges to similar shapes of a heat kernel, with attentions being allocated to structurally similar nodes in a very similar way as reported by [9] on barbell graphs. On the other hand, on heterophilic networks, MLP emphasizes on mid-high frequency range which contributes the most to performance, which would not be able to achieve using heat or pagerank kernels. We agree with you that the ChebNet can approximate high-frequency filter in some cases. However, there resides two key differences between these two models:
> > > * ChevNet aggregates (or normalizes) node features via a spectral filter, whereas our model uses wavelet basis to measure the similarity between nodes via attention (normalization/aggregation follows after this).
> > > * Our model uses adaptive filters via multi-head attention, where each head learns attention weights from a different spectral filter, whereas ChevNet uses the `same polynomial filter` over the entire network. Therefore, ChebNet can only use overfitted (if it happens - ChevNet has a very small number of parameters so it is unlikely) filters whereas our model can use appropriate filters, through the standard training-validation process.
> > >
> > > ### Complexity
> > > In our work, Eq.(3) is computed via Chebyshev polynomials, which means $\psi_{u,v}$ for a pair of nodes $(u,v)$ is only computed when they are linked within $p$-hop neighborhood. Combining this with sparse-matrix structure, we can achieve complexity $O(p\times|E|)$ for the attention part.
> > >
> > > Given the confusion, we have added some details of the Chebyshev polynomial approximation in Appendix A. Please take a look and let us know if further clarification is needed.
> > >
> > >
> > > [5] Thomas N. Kipf and Max Welling. “Semi-supervised  classification  withgraph convolutional networks” \
> > > [6] Fan Chung and Wenbo Zhao. “PageRank and random walks on graphs” 2010\
> > > [7] Fan Chung. “The heat kernel as the pagerank of a graph” 2007\
> > > [8] Johannes Klicpera, Stefan Weißenberger, and Stephan G ̈unnemann. “Dif-fusion Improves Graph Learning" 2019\
> > > [9] Claire Donnat et al. “Learning  structural  node  embeddings  via  diffusionwavelets” 2018

---

> ### Comment · Area_Chair1 · 2020-11-23
> **thanks for the response**
>
> Thank you for providing detailed response.
>
> R1, could you carefully read the responses by authors?
>
> In the mean time, I have quick question to authors: in figure 3, why did you just evaluate the computational efficiency of your methods? could you provide some comparisons against other baselines to support your statements?

---

> > ### Author Response · Authors · 2020-11-25
> > **Runtime comparison**
> >
> > Thanks for spending time reading the latest revision.
> >
> > Figure 3 was added for the purpose of demonstrating how the two sparsification tricks improve runtime efficiency of the model itself, which is why we didn't compare with other baselines. That said, we acknowledge an empirical comparison with other baselines would be useful. Given the time constraint, we hereby provide a simple comparison limited to two baselines. We will add the other benchmark results in later revisions.
> >
> > | Dataset | GAT | Geom-GCN | GNAN | GNAN-K |
> > | --- | --- | --- | --- | --- |
> > |Cora | 98ms | 172ms | 161ms | 95ms |
> > |Chame. | 71ms | 175ms | 114ms |  65ms|
> >
> > The above runtime results are the milliseconds per training epoch averaged over 500 epochs. Our model GNAN is slower than GAT but faster than Geom-GCN.
> > Note the reported runtimes of GNAN are larger than the ones in Figure 3 because we disabled optimizations using sparse matrix operations for a fair comparison with the two baselines. GNAN-K has the optimization turned on, where K=5.
> >
> > Results are obtained on  GeForce RTX 2080 Ti with 12G of GRAM.
> > For the baselines, we use the source code from:\\
> > GAT: https://github.com/Diego999/pyGAT\\
> > Geom-GCN: https://github.com/graphdml-uiuc-jlu/geom-gcn

---

### Official Review · AnonReviewer2 · 2020-10-27
**Official Blind Review #2**

**Rating:** 7
**Confidence:** 4

**Review:**

Main Idea

In this paper, the authors study the problem of GCN for disassortative graphs. The authors proposed the GNAN method to allow attention on distant nodes indeed of limiting to local neighbors. The authors generalized the idea of graph wavelet with MLP to generate the attention score and utilized it to generate multiple attention heads. The authors carried out experiments on several real-world networks (4 assortative and 3 disassortative) with comparison to several state-of-art GCN methods.

Strength:
The authors study a very interesting problem of GCN/graph embedding or disassortative graphs.
The proposed method is well motivated with solid theoretical motivation from graph wavelets. The proposed model is very intuitive generalization of graph wavelet methods.
The empirical evaluation is very thorough on seven networks with comparison to about 10 baselines of different kinds.

Weakness:
Though the authors mentioned the use of sparsification of attention for speed-up, however, it mentioned that t is set to zero. It is interesting to see how scalable the proposed method is as it needs to have global attention to possibly all nodes. An empirical comparison of running time would be very helpful.
 The authors only carry out experiments on three disassortative which are all very small. It would be interesting to see more experiments on disassortative graphs. Alternatively, it would be interesting to have an experiment on synthetic graphs where the \beta can be controlled and varied smoothly to see how it affects the performance of different algorithms.
The authors picked only node classification of evaluation tasks. It is interesting to see how the disassortative could impact other tasks like graph reconstruction and link prediction.

---

> ### Author Response · Authors · 2020-11-23
> **Response to AnonReviewer #2: more datasets and sparsification details**
>
> Thank you very much for the constructive feedback. We have revised our paper accordingly. Please find below our responses.
>
> ### (1) Attention sparsification
>
> Following your feedback, we have added further technical details (Section 3) and new experimental results (Section 4.2) in our revised paper. To gain a better understanding on how attention sparsity can affect efficiency, we have included two sparsification techniques in our experiments: (a) one is based on a threshold; and (b) the other is based on top-k sorting. As depicted in Figure 3 in the revised paper, both sparsification techniques can significantly improve runtime efficiency.
>
> ### (2) Datasets
>
> In the revised paper, we have added another larger disassortative network, Chameleon, which has 2,277 nodes and 36,101 edges.
> Our method GNAN performs the best across all the baselines on this dataset. We are also doing further experiments on other disassortative networks, and will report the results once they are available.
>
> Conducting experiments on synthetic graphs with a controllable $\beta$ is a great idea. We will look into how such graphs can be constructed and add experimental results later on. In the meantime, if you are aware of any existing work that generates such graphs, please let us know.
>
>
> ### (3) Evaluation tasks
>
> Node classification is commonly used to evaluate the model performance by state-of-the-art GNN methods. To have a fair and comprehensive comparison with state-of-the-art GNN methods, we thus benchmark our model against these methods on node classification in this work. We appreciate your suggestions to benchmark on other evaluation tasks such as graph reconstruction and link predication, as well as graph classification, and will  work on these tasks as a next step.

---

### Official Review · AnonReviewer4 · 2020-10-28
**This paper proposes a novel method for Graph Neural Networks with adaptive spectral filters that experimentally outeprform other GNN designs and has comparable performance with MLP in graphs having small local homophily.**

**Rating:** 7
**Confidence:** 1

**Review:**

I liked this paper quite a lot. Although this paper does not belong to my area of expertise, I was able to understand the paper clearly because of its lucid exposition. Experimentally, the authors show a novel GNN design with an attention module that has comparable performance to the MLP and outperforms other GNN designs. I believe that this will be a valuable contribution to many practical problems.

Unfortunately, this work does not have any theoretical results, and evaluating the experimental results is outside my range of expertise. Therefore I would like to defer this paper to my fellow reviewers.

---

> ### Author Response · Authors · 2020-11-23
> **Response to AnonReviewer 4: Thank you for the positive feedback**
>
> Thank you for the positive comments.  We are delighted to see that you are able to understand the paper, even though part of the paper is not in your area of expertise.  We believe it is important to make the manuscript accessible to a wide range of researchers in the community who may not necessarily have deep knowledge on the subject.  Please do not hesitate to let us know if you have any questions later on.

---

### Decision · Program_Chairs · 2021-01-07
**Final Decision**

**Decision:**

Reject

**Comment:**

This paper proposes a GNN that uses global attention based on graph wavelet transform for more flexible and data-dependent GNN feature aggregation without the assumption of local homophily.

Three reviewers gave conflicting opinions on this paper. The reviewer claiming rejection questioned the novelty of the paper and the complexity of the global attention mentioned in the paper. Even through the authors' responses and subsequent private discussions, concerns about complexity and novelty were not completely resolved.


Considering the authors' claim that the core contribution of this paper is to design fully learnable spectral filters without compromising computational efficiency, it is necessary to consider why it is meaningful to perform global attention based on graph wavelet transform in the first place. In terms of complexity, although the wavelet coefficient can be efficiently calculated using the Chebyshev polynomials mentioned by the authors, in the attention sparsification part, n log n is required **for each node** in sorting, resulting in complexity of n^2 or more. There may still be an advantage of complexity over using global attention in a message-passing architecture, but it will be necessary to clarify and verify that, given that the proposed method uses an approximation that limits global attention within K hops.

Also, this paper modifies the graph wavelet transform in graph theory, which requires a deeper discussion. For example, as the authors mentioned, the original wavelet coefficient psi_uv can be interpreted as the amount of energy that node v has received from node u in its local neighborhood. The psi_uv defined by the learnable filter as shown in Equation 3 has a different meaning from the original wavelet coefficient. There is insufficient insight as to whether it is justifiable to use this value as an attention coefficient.

Overall, the paper proposes potentially interesting ideas, but it seems to require further development for publication.